# Distinct Cellular Profiles of *Hif1a* and *Vegf* mRNA Localization in Microglia, Astrocytes and Neurons during a Period of Vascular Maturation in the Auditory Brainstem of Neonate Rats

**DOI:** 10.3390/brainsci11070944

**Published:** 2021-07-18

**Authors:** Daphne Chang, Quetanya Brown, Grace Tsui, Ye He, Jia Liu, Lingyan Shi, Adrián Rodríguez-Contreras

**Affiliations:** 1Center for Discovery and Innovation, Department of Biology, Institute for Ultrafast Spectroscopy and Lasers, City University of New York, City College, New York, NY 10031, USA; daphnec95@gmail.com (D.C.); quetanya.brown@gmail.com (Q.B.); graceeetsui@gmail.com (G.T.); 2Neuroscience Initiative, Advanced Science Research Center at the Graduate Center, City University of New York, New York, NY 10031, USA; yhe1@gc.cuny.edu (Y.H.); jliu1@gc.cuny.edu (J.L.); 3Department of Bioengineering, University of California San Diego, La Jolla, CA 92093, USA

**Keywords:** auditory brainstem, hypoxia, microglia, *Hif1a*, *Vegfa*

## Abstract

Defining the relationship between vascular development and the expression of hypoxia-inducible factors (*Hifs*) and vascular endothelial growth factor (*Vegf*) in the auditory brainstem is important to understand how tissue hypoxia caused by oxygen shortage contributes to sensory deficits in neonates. In this study, we used histology, molecular labeling, confocal microscopy and 3D image processing methods to test the hypothesis that significant maturation of the vascular bed in the medial nucleus of the trapezoid body (MNTB) occurs during the postnatal period that precedes hearing onset. Isolectin-B4 histochemistry experiments suggested that the MNTB vasculature becomes more elaborate between P5 and P10. When combined with a cell proliferation marker and immunohistochemistry, we found that vascular growth coincides with a switch in the localization of proliferating cells to perivascular locations, and an increase in the density of microglia within the MNTB. Furthermore, microglia were identified as perivascular cells with proliferative activity during the period of vascular maturation. Lastly, combined in situ hybridization and immunohistochemistry experiments showed distinct profiles of *Hif1a* and *Vegf* mRNA localization in microglia, astrocytes and MNTB principal neurons. These results suggest that different cells of the neuro-glio-vascular unit are likely targets of hypoxic insult in the auditory brainstem of neonate rats.

## 1. Introduction

Perinatal asphyxia is a major factor responsible for severe injury in preterm and term neonates. Human neonates who experience asphyxia may develop hearing loss and poor speech discrimination [1], while rats exposed to controlled anoxia at birth show deficits that are consistent with slower neural processing of sound [2]. However, the molecular factors and the cellular populations that are affected by oxygen shortage in the auditory brain of neonates have not been fully identified.

The vascular system is at the interface between an environmental oxygen challenge and the local tissue response. The vascular supply to the brain consists of blood vessels that carry out gas exchange, nutrient delivery, and immune support to meet local tissue metabolic requirements and maintain homeostasis [3]. Regulation of vascular development involves the synergy between signaling pathways activated by hypoxia-inducible transcription factors (*Hifs*) and vascular endothelial growth factor (*Vegf*). Expression of Hifs and Vegf in different cell types enables vascular growth, maintenance, or regression in response to physiological and pathological hypoxia [4,5,6,7]. Hence, defining the relationship between vascular development and the expression of Hifs and Vegf in the auditory brainstem may reveal clues to understand how tissue hypoxia contributes to the sensory deficits caused by oxygen shortage in neonates.

In this study we investigated different aspects of vascular development in the medial nucleus of the trapezoid body (MNTB), an auditory brainstem nucleus in the superior olivary complex (SOC) of mammals [8,9]. The MNTB consists of a population of principal neurons that are innervated by the calyces of Held, specialized glutamatergic synaptic terminals that originate from cochlear nucleus globular bushy neurons (Figure 1A) [8]. Recent studies showed significant changes in the development of MNTB astrocytes, oligodendrocytes and microglia during the postnatal period that precedes hearing onset, which in rats occurs around postnatal day 12 (P12) [10,11,12,13,14,15]. However, to the best of our knowledge, the development of the MNTB vasculature has not been characterized in any species. The main hypothesis of this study is that significant maturation of the MNTB vascular bed occurs during the postnatal period that precedes hearing onset (Figure 1B). If this is correct, we predict that Hifs and Vegf will be expressed in different cell types in the MNTB.

To test this hypothesis, we performed molecular labeling and histological studies in the MNTB of neonate rats between birth (P0) and P21. We found that the MNTB vasculature becomes more elaborate between P5 and P10. Vascular growth coincided with a switch in the perivascular localization of proliferating cells as well as an increase in the density of perivascular microglia, and distinct profiles of *Hif1a* and *Vegf* mRNA expression in microglia, astrocytes and MNTB principal neurons. Based on these results, we propose that due to their proximity to blood vessels and the expression profile of *Hif1a* and *Vegf*, different cells of the neuro-glio-vascular unit are likely targets of hypoxic insult in the auditory brainstem of neonate rats.

## 2. Materials and Methods

### 2.1. Animals

Experimental procedures were reviewed and approved by the Institutional Animal Care and Use Committee of the City College of New York. Wistar rat males and dams were obtained at postnatal age (P) 70 (Charles River, Catskill, NY, USA). Upon arrival to the animal facility, animals were housed for one week in same sex pairs for acclimation purposes. After acclimation, breeding trios consisting of one male and two females were housed together for one week. After the breeding period, males were removed from the study. Pregnant rats were housed in pairs for 18 days, and then housed individually in a new room until they gave birth. Pups were housed with their mothers from birth (P0) until P21. A total of 52 pups from either sex were used in this study.

### 2.2. Histology

Rat pups received an intraperitoneal injection of euthasol (100 mg/kg; Virbac, Westlake, TX, USA) and were monitored every minute for toe pinch reflex to determine complete sedation. Pups were perfused intracardially with 4% formaldehyde in 0.1 M phosphate buffer (PB; pH 7.4) using a syringe pump at cardiac output rate [16,17]. Brains were dissected, postfixed overnight and immersed sequentially in 10%, 20%, and 30% sucrose in 0.1 M PB for up to 48 h. After cryoprotection, 60–80 μm thick horizontal or coronal brainstem sections were obtained with a freezing stage microtome (American Optical, Southbridge, MA, USA) and used for free floating tissue-staining experiments. For combined in situ hybridization and immunohistochemistry experiments, 15–20 μm thick horizontal sections were obtained with a cryostat (Leica Biosystems, Buffalo Grove, IL, USA) and staining was performed on tissues mounted on glass slides. Two section planes were used for experiments. Horizontal 80 μm thick brainstem sections were used to visualize the caudal to rostral and medial to a lateral extent of the medial nucleus of the trapezoid body (MNTB) [11]. Coronal, 60–80 μm thick brainstem sections were used to visualize the medial to a lateral extent of the MNTB and other nuclei in the superior olivary complex.

### 2.3. Isolectin-B4 (IsoB4) Histochemistry

Staining procedures were modified from previous descriptions [11]. Briefly, brainstem sections containing the MNTB were selected and incubated in a blocking solution containing (in %) 1.0 goat serum and 0.4 Triton X-100 dissolved in 0.1 M PB for 30 min at room temperature (RT). Brainstem sections were transferred to a primary label solution containing (in %) 1.0 goat serum, 0.3 Triton X-100, and biotinylated IsoB4 (1:500; Vector Labs, Burlingame, CA, USA), incubated for 2 h at RT followed by overnight incubation at 4 °C. After IsoB4 incubation, brainstem sections were washed twice in 0.1 M PB and incubated for 2 h at RT in secondary label solution containing (in %): 0.02 Triton X-100 and Alexa Fluor-conjugated streptavidin (1:500; Thermo Fisher Scientific, Waltham, MA, USA) followed by overnight incubation at 4 °C, washed 4 times in 0.1 M PB and mounted onto glass slides using a mounting medium with DAPI as a counterstain (Fluorogel, Electron Microscopy Sciences, Hatfield, PA, USA).

### 2.4. 5-Ethynyl-2’-deoyuridine (EdU) Tissue Labeling

EdU histochemistry was used as a marker of cell proliferation, using protocols described previously [11]. Briefly, pups were anesthetized with 3% isoflurane dissolved in medical grade oxygen and delivered via a nose cone. Anesthetized pups received a single intraperitoneal injection of EdU at 160 μg per gram of body weight. Animals were processed for histology two hours after EdU injection.

### 2.5. Multiple Labeling Experiments

Double or triple label experiments were performed with the same basic IsoB4 staining protocol. For combined IsoB4 histochemistry and immunohistochemistry experiments, the following primary antibodies were added at 1:500 dilution in separate sets to the primary label solution containing IsoB4: guinea pig anti-Iba1 (Synaptic Systems, Goettingen, Germany; Cat. No. 234 004); guinea pig anti-NeuN (Millipore Sigma, Burlington, NJ, USA; Cat. No. ABN90PMI), or rabbit anti-S100β (Immunostar, Hudson, NY, USA; Cat. No. 22520). Appropriate Alexa Fluor dye-conjugated secondary antibodies raised in goat were added to the secondary label solution containing Alexa Fluor dye-conjugated streptavidin at 1:500 dilution. For combined IsoB4 histochemistry, immunohistochemistry and EdU histochemistry experiments, Edu staining was performed at the end of the combined IsoB4 histochemistry and immunohistochemistry procedure using the Click-iT EdU imaging kit (Thermo Fisher Scientific, Waltham, MA, USA) with Alexa Fluor-conjugated hydrazide according to manufacturer’s instructions.

### 2.6. RNAscope Fluorescence in situ Hybridization and Immunohistochemistry

Frozen blocks of samples from the ventral brainstem were serially cut using a cryostat at 15–20 μm thickness on superfrost plus slides and kept at −80 °C until further processing. An in situ hybridization screen was performed using RNAScope probes Rt-*Hif1a* (432281-C1), Rt-*Vegf* (315361-C2) and reagents from Advanced Cell Diagnostics, according to the manufacturer’s instructions. Combined in situ hybridization and immunohistochemistry was performed using *Hif1a* and *Vegf* RNAScope probes and select antibodies. Briefly, sections were first fixed in chilled 4% paraformaldehyde for 15 min at 4 °C, dehydrated in increasing gradients of ethanol baths and left to air dry for 5 min. Endogenous peroxidase activity was quenched with hydrogen peroxide reagent for 10 min, followed by antigen retrieval for 5 min in boiling buffer. Immunohistochemistry was performed afterwards using anti-Iba1 to detect microglia, anti-S100β to detect astrocytes, or anti-NeuN to detect neurons, followed by protease digestion for 30 min at 40 °C. The RNAScope probes were then hybridized for 2 h at 40 °C in a humidity-controlled oven (HybEZ II, ACDbio, Newark, CA, USA). Successive addition of amplifiers was performed using the proprietary AMP reagents, and the signal was visualized through probe-specific horseradish-peroxidase-based detection by signal amplification with Opal dyes (Opal 520, Opal 570 Perkin Elmer, Waltham, MA, USA) diluted 1:1500. Slides were then counterstained with DAPI and coverslipped with Prolong Gold Antifade (Thermo Fisher Scientiffic, Waltham, MA, USA) and kept at 4 °C until imaging.

### 2.7. Confocal Microscopy and Data Analysis

A Zeiss LSM510 or a Zeiss LSM800 microscope (Zeiss, Jena, Germany) was used to image labeled brain sections. Confocal z-stack images (8-bit, 512 × 512 pixels) were obtained using a 10× (NA = 0.25; air) or a 40× lens (NA = 1.4; oil immersion). In order to reduce bias and enhance reproducibility, the medial region of the caudal MNTB was selected for imaging [11]. Fields of view were 1280 × 1280 µm at 10× magnification, and 319.5 × 319.5 µm at 40× magnification. Multi-channel confocal z-stacks were set to detect the following fluorophores with distinct excitation/emission properties: DAPI (355/460 nm), Alexa Fluor 488 (500/520 nm), Alexa Fluor 594 (590/620 nm), and Alexa Fluor 647 (650/670 nm). Appropriate band pass filters and separate laser lines were used to excite samples sequentially in order to minimize fluorescence cross talk amongst different detection channels. Confocal z-stacks were imported into Volocity (Perkin Elmer, Waltham, MA, USA) or Imaris (Oxford instruments, Zurich, Switzerland) for visualization, 3D rendering and analysis using available image-segmentation and analysis routines [18,19,20]. Data were plotted with Igor Pro (Wavemetrics, Portland, OR, USA).

### 2.8. Statistics

All data are presented as median values, or as mean ± SD. All statistical tests were performed in Prism 6 (GraphPad, San Diego, CA, USA). All datasets were examined with the D’Agostino and Pearson omnibus normality test. When data sets passed the normality test, the one-way ANOVA and Tukey’s multiple comparisons test were used to compare measurements between three or more ages. When data sets did not pass the normality test, the Kruskal–Wallis and Dunn’s multiple comparisons test were used to compare measurements between three or more ages. The two-tailed Mann–Whitney test was used to compare ranks between two non-matched sample pairs. The two-tailed Wilcoxon signed rank test was used to compare matched sample pairs. The two-tailed Kolmogorov–Smirnov test was used to compare cumulative distributions. Statistical significance was determined for probability *p* < 0.05. *P* values were adjusted for multiple comparisons.

## 3. Results

### 3.1. Postnatal Development of the Vascular Bed in the Medial Nucleus of the Trapezoid Body (MNTB)

We used the histochemical marker isolectin-B4 (IsoB4) to label and quantify changes in the density of labeled blood vessels in horizontal sections from the ventral brainstem between birth (P0) and P20 (Figure 2A). Figure 2B–E shows low- and high- magnification views from confocal z-stack projections of IsoB4-labeled horizontal brainstem slices. Analysis of z-stacks at high-magnification showed that the fluorescence density medians were higher at P10, P15 and P20 compared to P0 and P5. These differences were statistically significant (Figure 2F; *n* = 65 slices from 3 pups per age group at P0, P5 and P20, and 4 pups per age group at P10 and P15; one-way ANOVA F(26.55), *p* < 0.0001). However, we noted that IsoB4 also labeled cellular profiles adjacent to blood vessels in slices from P10 pups but not in slices from P0 pups. Hence, to address a potential confound of using fluorescence intensity based methods to quantify vascular structures at different ages, the number of branch points in the vascular network was determined by observers blind to age group. A statistically significant difference in the branch point medians was confirmed between P10, P15 and P20 compared to P5, and between P15 and P20 compared to P0 (Figure 2G; one-way ANOVA F(9.886), *p* < 0.0001). Altogether, these results suggest that the vascular structure of the MNTB increases between P5 and P10. However, these results do not exclude the possibility that IsoB4 labeling reflects maturational changes of a stable volume of blood vessels during the age range tested.

### 3.2. Postnatal Increase of Microglia Cells in the Rat MNTB

Next, we carried out a series of experiments to identify the cellular profiles labeled by IsoB4 in the MNTB at different ages. First, we noted that IsoB4 has been used previously to label microglia cells in brain slices from P12 or P15-P27 rats and mice [21]. Therefore, we performed double labeling experiments with anti-Iba1 immunohistochemistry, which has been used previously to label microglia in the auditory brainstem of neonate mice [10]. Using this approach, it was possible to visualize the vasculature and microglia simultaneously in coronal brainstem slices at different ages (Figure 3A). There were two main findings. First, we confirmed that all anti-Iba1 labeled cells were labeled by IsoB4 [21], implicating that similar to the cerebral cortex, microglia are closely associated with blood vessels in the neonate brainstem [22]. Second, we found that microglia abundance changed in the MNTB at different ages. To illustrate the latter finding, Figure 3E–G shows exemplar low magnification views of IsoB4-labeling in the superior olivary complex (SOC) at P1, P5 and P10. Note that the density of anti-Iba1 labeled cells in the SOC was relatively low at P1 and P5 compared to P10. As a result, the outline of the different SOC nuclei could be clearly discerned in anti-Iba1 labeled tissue slices at P1 and P5 compared to P10. Based on this observation, we quantified the number of microglia in regions of interest (ROI) located inside (IN) or outside (OUT) the MNTB. We found that the numbers of anti-Iba1 labeled cells in IN and OUT ROI, respectively, were 2.2 ± 1.4, and 8.6 ± 3.7 at P1 (*p* = 0.0297), 7.1 ± 2.0, and 8.5 ± 2.0 at P5 (*p* > 0.9999), and 31.0 ± 5.8, and 21.0 ± 8.0 at P10 (*p* > 0.9999). Furthermore, statistically significant differences were obtained for all comparisons between P1 and P10, and between P5 and P10 (Figure 3K; *n* = 52 slices from 4 pups per age group; Kruskal–Wallis statistic (6, 104) = 86.29, *p* < 0.0001). Lastly, we calculated the microglia ratio, defined as the quotient of the number of anti-Iba1 labeled cells between IN and OUT ROI. Microglia ratios were 0.3 ± 0.2 at P1, 0.9 ± 0.3 at P5, and 1.7 ± 0.8 at P10, which showed statistically significantly differences from each other (Figure 3L; Kruskal–Wallis statistic (3, 52) = 35.81, *p* < 0.0001). Altogether, these results show that subtle changes in the number of MNTB microglia occur at P1, and that more robust changes in the number of MNTB microglia occur between P5 and P10. The latter period coincides with the time of vascular changes identified in Figure 2.

### 3.3. Postnatal Change in the Perivascular Localization of Proliferating Cells in the MNTB

Next, we considered that cell proliferation could be involved in the increase of microglia observed between P1 and P10. To examine cell proliferation in the MNTB vascular niche, we performed acute injections of the thymidine analog 5-ethynyl-2’-deoyuridine (EdU) and performed double label experiments with IsoB4 histochemistry (Figure 4). We identified three main groups of EdU-labeled cells in 3D rendering analyses of confocal z-stacks. Parenchymal (Pa) cells were localized away from IsoB4-labeled vessels (Figure 4A), perivascular (Pe) cells were observed in contact with the abluminal side of IsoB4-labeled vessels (Figure 4B), and vascular (V) cells were observed in the intraluminal side of IsoB4-labeled vessels (Figure 4A). Next, we compared the proportion of Pa, Pe and V cells at different ages. Pa cells comprised 64 ± 28% of EdU-labeled cells at P1, and decreased to 36 ± 13% and 34 ± 6% of EdU-labeled cells at P7 and P10, respectively (Figure 4C). Pe cells comprised 36 ± 13% of EdU-labeled cells at P1, and increased to 67 ± 15% and 66 ± 11% at P7 and P10, respectively (Figure 4D). Lastly, V cells comprised 1.4 ± 1.5% of EdU-labeled cells at P1, 2.3 ± 2.1% of labeled cells at P7 and 2.4 ± 3.1% of EdU-labeled cells at P10 (Figure 4E; *n* = 35 slices from 3 pups per age group; Kruskal–Wallis and Dunn’s multiple comparisons test (9, 156), *p* < 0.0001). These results show that a larger percent of EdU-labeled cells associate with perivascular locations at P7 and P10 compared to P1, and provide evidence that IsoB4-labeled blood vessels contain proliferating cells, which is consistent with the interpretation that new vessels form between P5 and P10.

### 3.4. Microglia Are Perivascular Cells with Proliferative Activity

Based on the previous analyses of microglia number and cell proliferation in the vascular niche, we examined in more detail the relationship between the cellular profiles labeled by IsoB4 histochemistry, anti-Iba1 immunohistochemistry and Edu histochemistry at P1 and P7 (Figure 5). First, we analyzed the distribution of anti-Iba1 labeled, and anti-Iba1 EdU double labeled cells in the local region of the MNTB. We found that at P1, the number of anti-Iba1 labeled cell bodies located in the MNTB was 7 ± 2 compared to 20 ± 4 located outside (Figure 5A; Wilcoxon matched-pairs signed rank test; *p* = 0.0039; *n* = 9 slices from 3 pups). In contrast, at P7 the number of anti-Iba1 labeled cell bodies located in the MNTB was 29 ± 11 compared to 27 ± 5 located outside (Figure 5B; *n* = 7 slices from 3 pups; Wilcoxon matched-pairs signed rank test; *p*=0.7813). Based on this data, we calculated a microglia ratio of 0.3 ± 0.1 at P1, and a microglia ratio of 1.2 ± 0.6 at P7 (Figure 5C; Mann–Whitney U = 4, *p* = 0.0017). Second, we found evidence of anti-Iba1 and EdU double labeled cells at P1 and P7 (indicated with white arrowheads in Figure 5A,B). We noted that all double labeled cells were located outside the MNTB outline. Although the range of anti-Iba1 and EdU double labeled cells was shorter (0 to 5%) at P7 compared to P1 (0 to 18%), the percent of double labeled cells was not statistically different between P1 (6.6 ± 5.9) and P7 (2.4 ± 1.5; Figure 5D; Mann–Whitney U = 13.5, *p* = 0.0567). This result shows that some Pe cells are perivascular microglia.

Lastly, we used image processing and segmentation tools to quantify the vessel coverage by anti-Iba1 labeled cells at P1 and P7 (Figure 6). This procedure allowed us to obtain an independent measure of the increase in anti-Iba1 labeled cells in the MNTB. The increase in anti-Iba1 labeled cells corresponded to an increase in the percent vessel coverage at P7 compared to P1 (Figure 6C). In sum, the results in Figure 5 and Figure 6 confirm that microglia distribution changes during development, and provide additional support to the interpretation that microglia are perivascular cells with proliferative activity.

### 3.5. Localization Profiles of Hif1a and Vegfa mRNAs in the MNTB of Neonate Rats

A recent study documented the expression profiles of genes involved in hypoxia-dependent signaling in different regions of the neonate rat auditory system [15]. Motivated by this finding a multi-label in situ hybridization screen was performed to analyze the localization of *Hif1a* and *Vegf* mRNAs in the brainstem of neonate rats (*n*= 4 rats at P7). We found that *Hif1a* and *Vegf* mRNA probes labeled distinct clusters throughout the brainstem, suggesting that they were expressed in different cells, or in different proportions in multiple cell types.

To examine the localization of mRNA probes in different cell types, we used a combination of in situ hybridization with *Hif1a* and *Vegf* mRNA probes, and immunohistochemistry probes for microglia, astrocytes, and neurons in four additional P7 rat pups (Figure 7). Figure 7A–C shows 3D renderings from an exemplar confocal stack of triple stained tissue with anti-Iba1 immunohistochemistry (shown in blue) and probes against *Hif1a* and *Vegf* mRNAs (shown in green and orange, respectively). Figure 7A shows the profile of anti-Iba1 label alone, which can be compared with Figure 7B,C, in which the fluorescence of *Hif1a* and *Vegf* signals are shown with respect to the microglia marker, respectively. Note that anti-Iba1 labeled cells were relatively small, elongated and scattered throughout the tissue, while *Hif1a* fluorescence occurred in ovoid shaped particle clusters (Figure 7B). In contrast, the *Vegf* probe fluorescence formed smaller and more compact particle clusters (Figure 7C). To get an estimate of the number of mRNA clusters per anti-Iba1 labeled cell we used image segmentation and co-registration analyses. Figure 7D shows cumulative histograms of the number of mRNA particles per Iba1-labeled cell. Although cumulative histograms were statistically significantly different between *Hif1a* and *Vegf* mRNA probes (*n* = 52 cells; Kolmogorov–Smirnov statistic = 0.4038; *p* = 0.0004), the relative number of mRNA particles per cell was relatively low.

Next, we examined the localization profile of mRNA probes and anti-S100β labeled astrocytes (Figure 7E–G). We found that *Hif1a* probe fluorescence did not show an obvious overlap with this population of cells (Figure 7F). In contrast, the *Vegf* probe labeled particle clusters that showed a considerable match with anti-S100β labeled cells (Figure 7G). The cumulative histograms showed a larger range of *Vegf* mRNA probe labeled particles compared to *Hif1a* mRNA probe labeled particles in anti-S100β labeled cells (Figure 7H; *n* = 335 cells; Kolmogorov–Smirnov statistic = 0.4365; *p* < 0.0001).

Lastly, we examined the localization profile of mRNA probes in neurons identified with anti-NeuN immunohistochemistry (Figure 7I–K). We found that anti-NeuN labeled cells had a noticeable match with Hif1a mRNA probe fluorescent particle clusters (Figure 7J). In contrast, Vegf mRNA probe fluorescent particle clusters were observed adjacent to anti-NeuN labeled cells (Figure 7K). The cumulative histograms showed a larger range of *Hif1a* mRNA probe labeled particles compared to *Vegf* mRNA probe labeled particles in anti-NeuN labeled cells (Figure 7L; in = 961 cells; Kolmogorov–Smirnov statistic = 0.4810; *p* < 0.0001).

## 4. Discussion

During development, the vascular network undergoes different stages of growth. First, through a process of vasculogenesis, in which blood vessels form de novo, and thereafter through a process of angiogenesis, in which blood vessels grow from an existing vascular network. Angiogenesis is one of the processes under transcriptional regulation by hypoxia-inducible factors (Hifs) and vascular endothelial growth factor (Vegf) in the CNS of neonate rodents [4,5,7]. In this study, we performed a molecular labeling approach to characterize for the first time the development of the vascular bed in the medial nucleus of the trapezoid body (MNTB). The cell proliferation marker EdU labeled cells located inside IsoB4-labeled blood vessels (V cells in Figure 4), which supports the interpretation that angiogenesis contributes to the growth of the vascular bed in the MNTB, as others have proposed for the cerebral cortex using similar methods [23,24]. It has been noted that proliferation of endothelial cells may have consequences for the development of vascular permeability [25]. Our preliminary studies of fluorescent dextran imaging in the rat auditory brainstem suggest that there is a decrease in the permeability of microcapillaries after P8 [26]. Future preclinical studies could examine in more detail the relationship between proliferation of endothelial cells and the expression of molecular factors involved in the development of the blood–brain barrier under physiological and pathological contexts.

This study is the first to show the localization of *Hif1a* and *Vegf* mRNAs in microglia, astrocytes and principal neurons of any auditory brainstem nucleus in neonates. In the auditory system, *Hif1a* mRNA expression had been measured in tissue samples of the cochlea [27], and our recent study measured *Hif1a* expression in the cochlear nucleus, the pons, the inferior colliculus, and the primary auditory cortex of neonate rats [15]. Hence, it is plausible that during an environmental oxygen shortage event such as in perinatal asphyxia, vascular development in the auditory brainstem may be affected due to the activity of the hypoxia-sensitive signaling pathway and its synergy with the Vegf signaling pathway. A limitation of this study is that we did not address the expression of *Hif1a* and *Vegf* in endothelial cells. However, the data suggest that MNTB astrocytes may be an important source of Vegf. Future studies could use additional endothelial cell, astrocyte and microglia markers such as CD31, aquaporin-4 and TMEM119 to examine in more detail *Hif1a*, *Vegf*, and *Vegf* receptor expression during the developmental period identified in this study [23,28,29,30].

Another important finding of this study is the identification of microglia cells as perivascular cells that proliferate and increase in numbers during the same stage of vascular development in the superior olivary complex. This result supports the hypothesis that microglia promote angiogenesis, as previous studies have shown by genetic and pharmacological manipulation of cells of monocytic and hematopoietic lineages that affect blood vessel development in mice (reviewed in [31]). In our previous work, we found evidence that proliferation of MNTB astrocytes occurred during the perinatal period that precedes hearing onset [11]. Other groups have reported proliferation of oligodendrocytes in the mouse MNTB [14]. Future studies could evaluate the localization of proliferating astrocytes and oligodendrocytes with respect to the vasculature. A limitation of the present study is that we could not evaluate if cell migration contributes to the observed changes in the number and distribution of microglia cells in the auditory brainstem. Future studies could evaluate cellular dynamics using time-lapse microscopy [32]. Relevant to this point, we would like to note that the rodent auditory system has high metabolic requirements that mature during the perinatal stage, which overlaps with the onset of hearing. Auditory processing relies on fast and reliable neurotransmission, which has been linked to the high metabolic demand of auditory neurons [33]. The following features make the auditory system particularly sensitive to tissue hypoxia: auditory brain regions have the highest levels of metabolic activity which has been shown in studies with isotopes [34]; hypoxia potentiates noise-induced hearing loss and auditory periphery damage [35], and the period preceding the onset of hearing is characterized by increased expression of metabolic markers and mitochondria abundance as shown in recent developmental studies [36,37]. We propose that the perinatal period that precedes hearing onset represents a period of high sensitivity to hypoxia caused by PA. Alternatively, the severity of PA in the auditory system could depend on whether the hypoxic insult occurs before or after hearing onset. Differences in the timeline of auditory development between different species pose a challenge to study vascular development in humans. However, future studies could take advantage of novel experimental platforms that allow controlled growth of complex human source tissue models to perform dynamic and high-spatial resolution analyses [38,39,40].

## 5. Conclusions

Based on the results of this study, we conclude that significant maturation of the vascular bed in the medial nucleus of the trapezoid body occurs during the postnatal period that precedes hearing onset.

## Figures and Tables

**Figure 1 brainsci-11-00944-f001:**
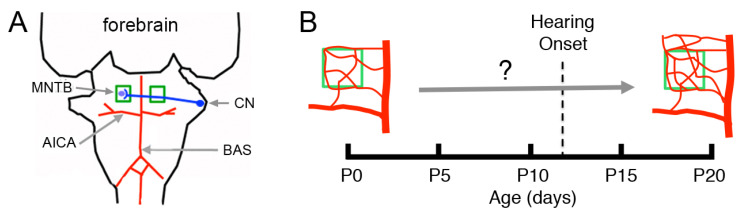
Main vascular supply to the ventral brainstem and hypothesis. (**A**) The anterior inferior cerebellar artery (AICA) is a main branch of the basilar artery (BAS). The BAS and AICA branch further to form the vascular beds that supply the brainstem. Neurons in the cochlear nucleus (CN, drawn in blue) connect synaptically with neurons in the contralateral medial nucleus of the trapezoid body (MNTB, enclosed in green squares). Little is known about the development of the MNTB vasculature. (**B**) It is hypothesized that the vasculature of the MNTB matures during the postnatal period before the onset of hearing.

**Figure 2 brainsci-11-00944-f002:**
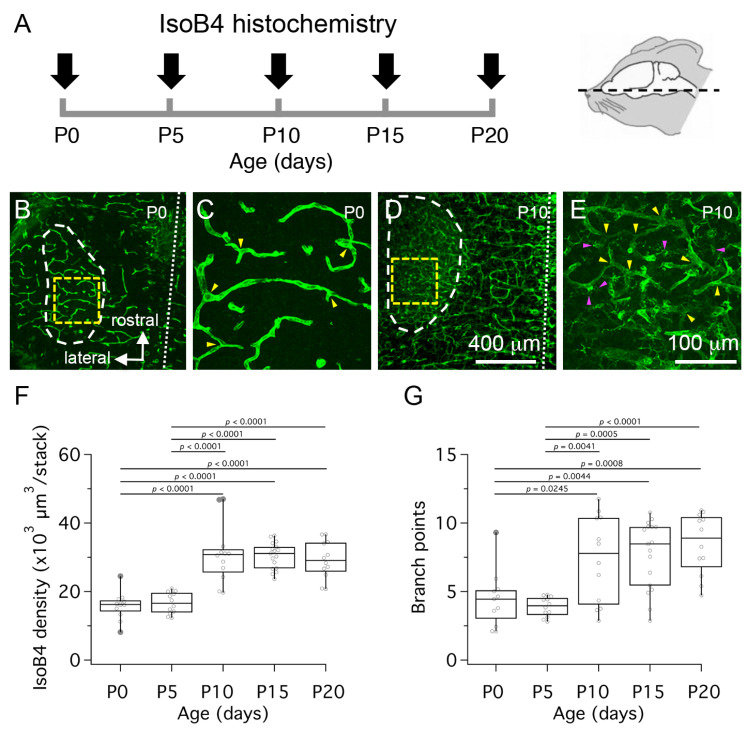
Postnatal development of the medial nucleus of the trapezoid body (MNTB) vascular bed. (**A**) To visualize the MNTB vascular bed, horizontal tissue sections from rat pups were processed for isolectin-B4 (IsoB4) histochemistry at different ages as indicated by black arrows. (**B**) Low magnification confocal z-projection view shows the MNTB outline (white dashed line) in a P0 brainstem slice stained with IsoB4 histochemistry. (**C**) High magnification confocal z-projection view of IsoB4 labeled blood vessels in the MNTB of a P0 brainstem slice. (**D**) Low magnification confocal z-projection view shows the MNTB outline (white dashed line) in a P10 brainstem slice stained with IsoB4 histochemistry. (**E**) High magnification view of IsoB4 labeled blood vessels in the MNTB of a P10 brainstem slice. (**F**) Quantification of IsoB4 fluorescence density at different ages. (**G**) Quantification of vascular branch points at different ages. Yellow dashed squares in panels (**B**,**D**) represent regions of interest examined at high magnification in panels C and E. White dotted lines in (**B**,**D**) represent the midline. Yellow arrowheads in (**C**,**E**) indicate vascular branch points. Magenta arrowheads in E indicate cellular profiles labeled with IsoB4 histochemistry. Scale bar in (**D**) applies to (**B**). Scale bar in (**E**) applies to (**C**). *P* values are corrected for multiple comparisons. Only statistically significant *p* values are printed.

**Figure 3 brainsci-11-00944-f003:**
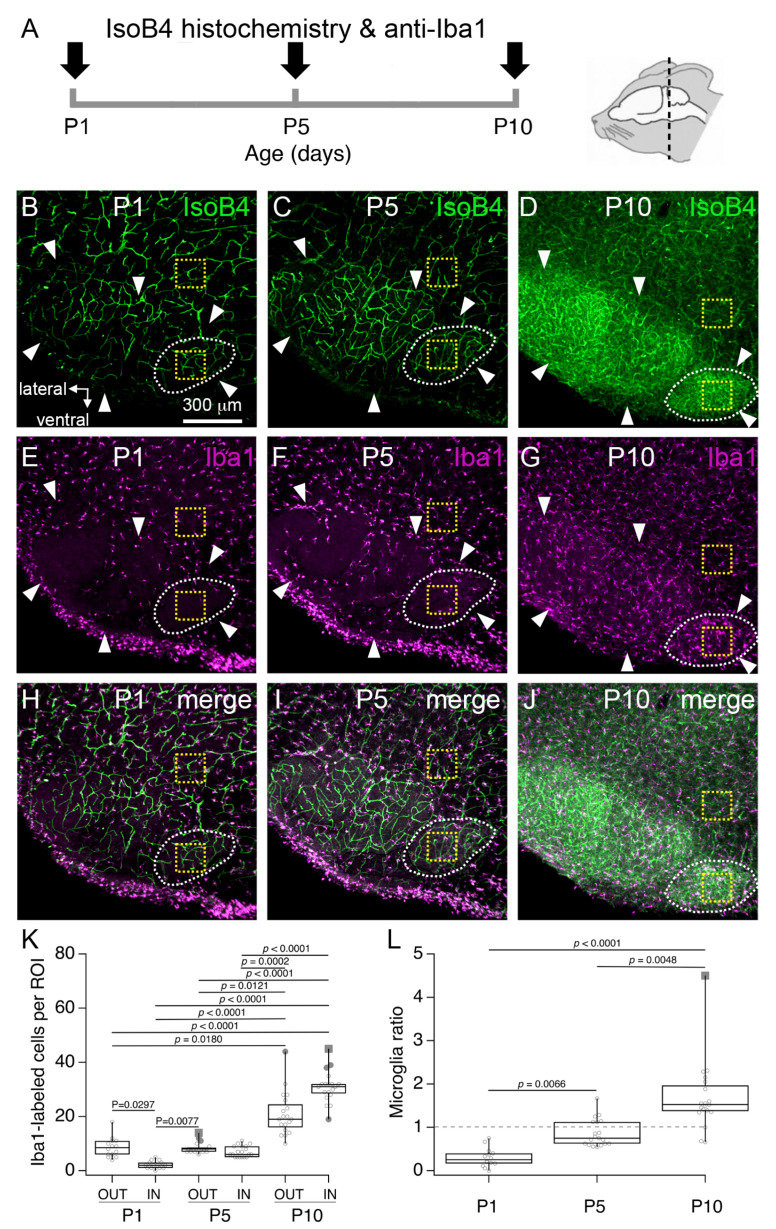
Changes in microglia number in the rat superior olivary complex (SOC). (**A**) Experiment outline. (**B**–**D****)** Exemplar low-magnification confocal z-stacks from P1, P5 and P10 brainstem sections show the profile of IsoB4 labeling in the SOC. (**E**–**G**) Anti-iba1 immunohistochemistry in the same tissue sections. (**H**–**J**) Merged views of IsoB4 and anti-Iba1 labeling. White arrows indicate the outline of the SOC. Dashed white lines indicate the outline of the MNTB. Yellow dashed squares indicate regions of interest (ROI) inside (IN) or outside (OUT) the MNTB. Scale bar in (**B**) applies to (**C**–**J**). (**K**) Number of anti-Iba1 labeled cells in IN ROI and OUT ROI at different ages. (**L**) Microglia ratio at different postnatal ages.

**Figure 4 brainsci-11-00944-f004:**
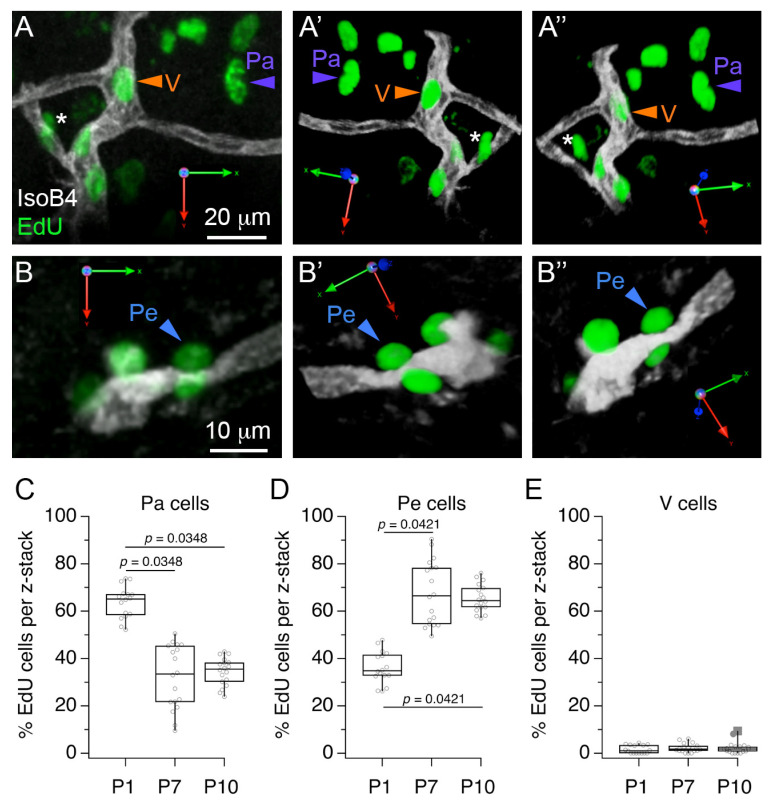
Classification of proliferative cells in the medial nucleus of the trapezoid body (MNTB) vascular niche. (**A**) Exemplar 3D rendering of IsoB4 and EdU double-labeled profiles illustrate the location of parenchymal (Pa) and vascular (V) EdU-labeled cells with respect to IsoB4-labeled vessels. Panels (**A’**,**A”**) indicate different rotation viewpoints. The asterisk indicates a parenchymal cell close to a vessel branch point. (**B**) Exemplar 3D rendering illustrates perivascular (Pe) EdU-labeled cells with respect to IsoB4-labeled vessels. (**B’**,**B”**) indicate different rotation viewpoints. (**C**) Percent Edu-labeled cells in the Pa category. (**D**) Percent EdU-labeled cells in the Pe category. (**E**) Percent EdU cells in the V category. Scale bars in (**A**,**B**) apply to (**A’,A”**,**B’**,**B”**), respectively.

**Figure 5 brainsci-11-00944-f005:**
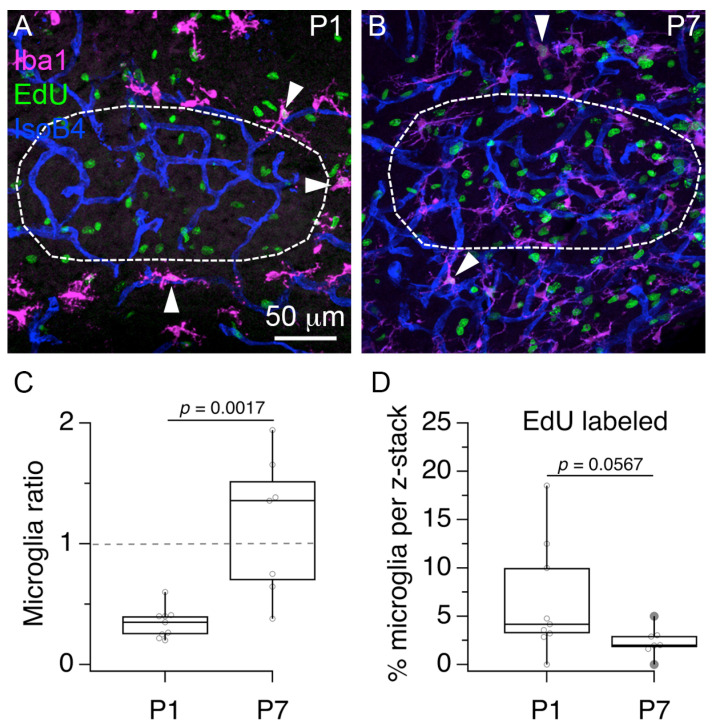
Visualization of proliferating microglia in the MNTB vascular niche. (**A**) Exemplar 3D rendering of triple labeled brainstem section at P1 shows the MNTB outline (white dashed line) and the localization of anti-Iba1 labeled cells (magenta), Edu-labeled cells (green) and IsoB4-labeled blood vessels (blue). (**B**) Exemplar 3D rendering of triple labeled tissue at P7. White arrowheads in A and B indicate anti-Iba1 and EdU double labeled cells. (**C**) Microglia ratio at P1 and P7. (**D**), Percent of anti-Iba1 labeled cells that co-labeled with EdU at P1 and P7.

**Figure 6 brainsci-11-00944-f006:**
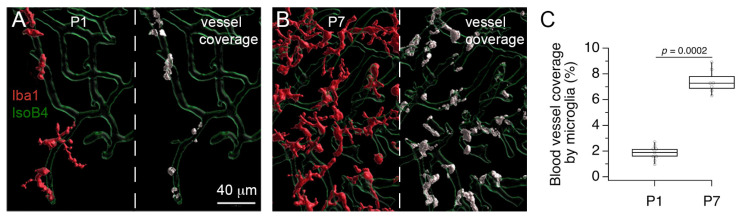
Increase in microglia vessel coverage in the MNTB. (**A**) 3D rendering after image segmentation shows microglia (maroon) and vessel (green) profiles in the MNTB at P1 (left). Coverage voxels between microglia and vessels are shown in white (right). (**B)** 3D rendering shows microglia and vessel profiles in the MNTB at P7 (left). Note the numerous covearge voxels between microglia and vessels (right). (**C**) Percent microglia vessel coverage at P1 and P7.

**Figure 7 brainsci-11-00944-f007:**
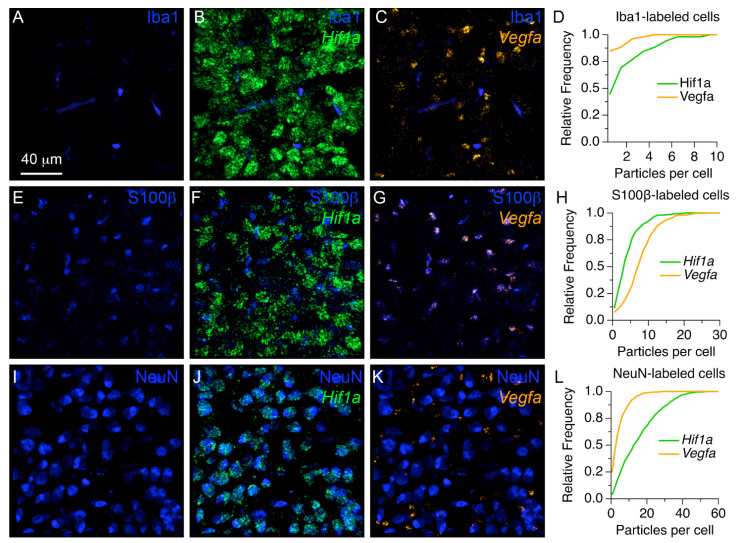
Localization of *Hif1a* and *Vegf* mRNAs in MNTB cells at P7. (**A**–**C**) Exemplar 3D rendering views show the localization of anti-Iba1 labeled cells (**A**) with Hif1a (**B**) or Vegf (**C**) mRNAs. (**D**) Cumulative histograms of the number of mRNA particles per anti-Iba1 labeled cell. (**E**–**G**) Exemplar 3D renderings show the localization of S100b-labeled cells (**E**) with respect to *Hif1a* (**F**) or *Vegf* (**G**) mRNAs. Note the obvious overlap of Vegf signal with S100b-labeled cells. (**H**) Cumulative histograms of the number of mRNA particles per anti-S100b labeled cell. (**I**–**K**) Exemplar 3D renderings of anti-NeuN labeled cells in the MNTB (**I**), with respect to Hif1a (**J**) or Vegf (**K**) mRNAs. Note the overlap between *Hif1a* signal and anti-NeuN labeling. (**L**) Cumulative histograms of the number of mRNA particles per anti-NeuN labeled cell. Scale bar in (**A**) applies to (**B**,**C**,**E**–**G,I**–**K**).

## Data Availability

Data reported in this study is available upon reasonable request.

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
