# Peer review of "Distinct Cellular Profiles of Hif1a and Vegf mRNA Localization in Microglia, Astrocytes and Neurons during a Period of Vascular Maturation in the Auditory Brainstem of Neonate Rats"

_brainsci, 2021, doi:10.3390/brainsci11070944_

Round 1

Reviewer 1 Report

Chang et al., presented the maturation of the medial nucleus of the trapezoid body (MNTB) in cellular levels and showed the vulnerability of MNTB for hypoxia using Hifs and Vegf markers in the auditory brainstem nucleus.

Following are my concerns;

Statistics: Authors could consider testing the significance difference between three or more ages using most common Turkey’s or Bonferroni’s test instead Dunn’s multiple comparisons test.

Fig.4, Authors have demonstrated a nice representation of increased number of Edu positive cells upon development; however, based on the Fig.4B (all the three panels), Pe pointed cells mostly look like non-endothelial cells. It would be helpful to understand very clearly if authors could perform double labelling of Edu against other endothelial cell markers (ex. CD31)  to validate their point here.

What is the status of endothelial specific Hif1a and Vegfa expression upon development? Authors could discuss.

Are there any changes in the expression levels of Hif1a and Vegfa in endothelial cells at the tested time points (P0->P20) ?

Does the endothelial cell proliferation strength the blood-brain-barrier(bbb) integrity?

Could these findings be noted for future preclinical studies regarding postnatal development related disorders, as well as designing drugs to by-pass the BBB?

Author Response

We would like to thank reviewer 1 for the comments and suggestions. Below is our reply. See also attached file with track changes on. 

  1. Statistics: Authors could consider testing the significance difference between three or more ages using most common Turkey’s or Bonferroni’s test instead Dunn’s multiple comparisons test.

We thank the reviewer for the suggestion. We used the D'Agostino normality test to determine if the ANOVA/Tukey or a non-parametric Kruskal-Wallis/Dunn test should be used. From all data sets with multiple comparisons, only the dataset in figure 1 passed the normality test. We have updated the text in the statistics section on page 4 and the results section on page 5. We updated P values in Figure 2F and Figure 2G. These changes did not affect the results in Figure 2 and the conclusions of the study.

2. Fig.4, Authors have demonstrated a nice representation of increased number of Edu positive cells upon development; however, based on the Fig.4B (all the three panels), Pe pointed cells mostly look like non-endothelial cells. It would be helpful to understand very clearly if authors could perform double labelling of Edu against other endothelial cell markers (ex. CD31)  to validate their point here.

Thank you for the comment and suggestion. We agree that Pe cells are unlikely to be endothelial cells. In fact, the identity of other EdU labeled cells is also puzzling. For example, V cells which are located inside vessels (intraluminal localization) are better candidates for proliferating endothelial cells compared to Pe cells, which are located around blood vessels (abluminal localization). In this study we confirmed that some Pe cells were labeled with anti-Iba1 immunohistochemistry, a marker of microglia.

In order to clarify our interpretation of the increase in Pe cells during development, we have made the following changes in the manuscript. First, to avoid confusion, we deleted the last sentence on page 8 ", and provide evidence that IsoB4-labeled blood vessels contain EdU-labeled cells, which is consistent with the interpretation that new vessels form between P5 and P10."; second, we stress the evidence of anti-Iba1 and EdU double labeled cells on page 9, and stress the main finding that some Pe cells are perivascular microglia. Lastly, we clarify in the discussion section that localization of cell proliferation markers in V cells inside IsoB4-labeled blood vessels support the interpretation that angiogenesis contributes to the growth of the MNTB vascular bed, and that future studies could use specific markers to analyze cell proliferation and gene expression in endothelial cells (second paragraph of page 12; see our answer to points 3 and 4 below). 

3. What is the status of endothelial specific Hif1a and Vegfa expression upon development? Authors could discuss.

This is a great question, and we acknowledge it as a limitation of our study. See second paragraph on page 12.

4. Are there any changes in the expression levels of Hif1a and Vegfa in endothelial cells at the tested time points (P0->P20) ?

See our response to point 3 above.

5. Does the endothelial cell proliferation strength the blood-brain-barrier(bbb) integrity? Could these findings be noted for future preclinical studies regarding postnatal development related disorders, as well as designing drugs to by-pass the BBB?

We address this issue in the first paragraph of the discussion, where we cite a review on this topic and our preliminary studies on blood brain barrier permeability. See new references 25 and 26. See also our response to point 5 from reviewer 2 (last paragraph of discussion section.

Reviewer 2 Report

Dear Editor,

The manuscript by Chang et al. reports that vascular growth coincides with a switch in the localization of proliferating cells to perivascular locations, and an increase in the density of microglia within the medial nucleus of the trapezoid body. These results suggest that that different cells of the neuro-glio- vascular unit are likely targets of hypoxic insult in the auditory brainstem of neonate rats.

The design of the study and the technical quality of the work look convincing and results can be of general interest. The manuscript is well-written and easy to follow and authors have implemented the right statistical approach. However, there is a number of major and minor points that would need to be addressed in order to improve the quality of this paper before it can be recognized:

Major:

A number of technical details are missing:

-Imaging was an essential aspect of this manuscript. Author needs to provide more details such as how many FOVs have been taken and what are their measures to minimize biases, and how they have excluded any possible interference from background signals in order to enhance the reproducibility of the presented data.

-In 2.2 “Rat pups were injected with an overdose of euthasol (Virbac)”. Authors need to specify the dose and comment on the rationale behind the selection of this particular dose and whether its effect is time- and/or concentration- dependent.

-In contrast to the current practice, astrocytes are not all necessarily S100B and GFAP positive. Authors need to discuss considering other biomarkers such as AQP4 which has been nicely shown by the work of Kitchen et al. Cell 2020 and this can be used in future studies. Reference to be included:

https://pubmed.ncbi.nlm.nih.gov/32413299/

Minor:

-Vegfa and vegf have been interchangeably throughout the manuscript. Authors need to correct this.

- Author needs to briefly discuss future directions following towards the end of their discussion and conclusion. This could include, but not limit to, the use of humanized self-organized models, organoids, 3D cultures and human organ-on-a-chip platforms especially those which are amenable for advanced imaging such as TEM and expansion microscopy since they enable real-time monitoring of cellular mechanisms. References to be included:

https://pubmed.ncbi.nlm.nih.gov/32640930/

https://pubmed.ncbi.nlm.nih.gov/33117784/

https://pubmed.ncbi.nlm.nih.gov/32864586/

Best.

Author Response

We would like to thank reviewer 2 for the comments and suggestions. Below is our reply. See also attached file with track changes on. 

1.-Imaging was an essential aspect of this manuscript. Author needs to provide more details such as how many FOVs have been taken and what are their measures to minimize biases, and how they have excluded any possible interference from background signals in order to enhance the reproducibility of the presented data.

Thank you for your comment. We have updated the methods section to clarify the selection material for imaging, and the measures to avoid artifacts in multiple-labeling experiments. See updated section 2.7 on page 4. Note that the number of slices represents the number of fields of view (FOV) used for analysis. This information is available in the results section.

2-In 2.2 “Rat pups were injected with an overdose of euthasol (Virbac)”. Authors need to specify the dose and comment on the rationale behind the selection of this particular dose and whether its effect is time- and/or concentration- dependent.

We clarify the dose of euthasol, a pentobarbital-like anesthetic approved by IACUC at our institution, and indicate the criteria used to determine its effect in time. See updated text of section 2.2 on page 3.

3-In contrast to the current practice, astrocytes are not all necessarily S100B and GFAP positive. Authors need to discuss considering other biomarkers such as AQP4 which has been nicely shown by the work of Kitchen et al. Cell 2020 and this can be used in future studies. Reference to be included:
https://pubmed.ncbi.nlm.nih.gov/32413299/

We agree that S100b is not a universal marker of astrocytes, but based on our previous studies (Saliu et al., 2014) it provided a marker for non-neuronal cell bodies. We updated the manuscript to indicate that future studies can use endothelial cell and astrocyte markers such as CD31 and aquaporin-4, including the relevant references. See second paragraph of discussion on page 12.

Minor:
4-Vegfa and vegf have been interchangeably throughout the manuscript. Authors need to correct this.

We have corrected this issue, and we now use Vegf throughout the manuscript.

5- Author needs to briefly discuss future directions following towards the end of their discussion and conclusion. This could include, but not limit to, the use of humanized self-organized models, organoids, 3D cultures and human organ-on-a-chip platforms especially those which are amenable for advanced imaging such as TEM and expansion microscopy since they enable real-time monitoring of cellular mechanisms. References to be included:
https://pubmed.ncbi.nlm.nih.gov/32640930/
https://pubmed.ncbi.nlm.nih.gov/33117784/
https://pubmed.ncbi.nlm.nih.gov/32864586/

Thank you for the suggestion. We include this information in the last paragraph of the discussion section on page 14 and added the suggested references.

Reviewer 3 Report

Brainsci-1262546-peer-review-v1

Summary: The authors have demonstrated that the maturation of vascular bed in the MNTB occurs between postnatal days P5 and P10 before the onset of hearing in Wistar rats. Such maturation is associated with cell proliferation in the perivascular regions and an increase in the density of microglia within the MNTB. They have also reported distinct cellular expression of angiogenic markers such as Hif1a and Vegfa in microglia, astrocytes and principal neurons of MNTB. These studies have delineated a relationship between vascular development and Hif1a-Vegfa pathway in the developing auditory brainstem.

Comments

1) Figure 2. It will increase the rigor of the study if the authors also analyze the volume of blood vessels and report if there are any differences or none during postnatal development.

2) Figure 3: 1) Include higher magnification showing juxtaposition of Iba1 positive cells with IsoB4 positive blood vessels. 2) Does this juxtaposition stabilize or decline after the onset of hearing once the maturation of vascular beds is over? 3) In addition to microglia, Iba1 also labels macrophages and perivascular macrophages (PVMs), hence include selective microglial markers such as P2ry12 or Tmem119.

3) Figure 3 E-G, The MNTB seems to be colonized by Iba1 positive cells beginning P5. What is the source of these cells? Could it be extravasation of blood circulating macrophages into the brain, or from the ventricles, or microglia migrating from parenchyma into and around the developing blood vessels in the MNTB? It looks like that the density of the Iba1 positive cells in the ventral SOC decreases from P5 to P10. Could these cells be migrating inside the SOC and into the MNTB? Include a discussion in the manuscript.

4) Figure 5A-C is repetitive information of what has been reported in Figure 3. Either remove this data or consolidate it with Figure 3.

5) As per data reported in Figure 5C, it doesn’t support the interpretation that microglia are the main perivascular cells with proliferative capacity inside the MNTB region as indicated in line 383 under discussion. In addition, the % of IBA1 and EdU double positive cells declines from P1 to P7. Perform double labeling for pericytes (NG2) or astrocytes or oligodendrocytes (GFAP or S-100ß) and EdU to determine if pericytes or astrocytes/oligodendrocytes are the main perivascular cells undergoing proliferation.

6) Figure 7A, immunolabeling of IBA-1 doesn’t match with previous figures. Please provide a representative figure. In addition, there doesn’t seem to be any significant overlap or co-localization of either Hif1a or Vegfa in IBA1 labeled cells. At least, it not evident from the representative images provided. Rather, it is evident in figures 7 E-K, that Hif1a is abundant in NeuN positive principal neurons of MNTB while Vegfa is abundant in S-100ß positive astrocytes/oligodendrocytes.

7) It will be worth discussing in the manuscript about the known roles of macrophages/microglia in vasculature development and angiogenesis in the brain and other tissues.  

Minor concerns

  1. Line 39: it is an incomplete sentence.
  2. It is unclear whether both male and female or either male or female Wistar rats were used in the study?

Author Response

Brainsci-1262546-peer-review-v1

Summary: The authors have demonstrated that the maturation of vascular bed in the MNTB occurs between postnatal days P5 and P10 before the onset of hearing in Wistar rats. Such maturation is associated with cell proliferation in the perivascular regions and an increase in the density of microglia within the MNTB. They have also reported distinct cellular expression of angiogenic markers such as Hif1a and Vegfa in microglia, astrocytes and principal neurons of MNTB. These studies have delineated a relationship between vascular development and Hif1a-Vegfa pathway in the developing auditory brainstem.

Comments

1) Figure 2. It will increase the rigor of the study if the authors also analyze the volume of blood vessels and report if there are any differences or none during postnatal development.

Respone: We thank the author for the suggestion of quantifying the volume of blood vessels.  To clarify, panel F in figure 2 shows the fluorescence volume in units of micrometers^3 per stack. The label indicates IsoB4 density because the fluorescence volume was normalized per stack.

2) Figure 3: 1) Include higher magnification showing juxtaposition of Iba1 positive cells with IsoB4 positive blood vessels. 2) Does this juxtaposition stabilize or decline after the onset of hearing once the maturation of vascular beds is over? 3) In addition to microglia, Iba1 also labels macrophages and perivascular macrophages (PVMs), hence include selective microglial markers such as P2ry12 or Tmem119.

Response: Thank you for the comment and request. 1) The main message of Figure 3 is to document changes in the number of Iba1-labeled cells in regions of interest in the ventral brainstem, hence the title “Changes in microglia number in the rat superior olivary complex (SOC)”. 2) The microglia vessel interaction was analyzed in Figure 6 at P1 and P7. We did not analyze this interaction at other ages. 3) We chose to use Iba1 as a microglia marker based on previous developmental studies in the SOC of mice (Dinh et al., 2014). Our study in rats confirmed the results in mice and also provides new data on changes in the distribution of Iba1 labeled cells in this brainstem region. Your suggestion to screen two additional antibodies of microglia markers is interesting but we consider that it is beyond the main scope of our study. Please see the updated second paragraph lines 422-425 in the discussion that states the need of future studies to use a more comprehensive set of markers to evaluate cellular changes in the auditory brainstem.

3) Figure 3 E-G, The MNTB seems to be colonized by Iba1 positive cells beginning P5. What is the source of these cells? Could it be extravasation of blood circulating macrophages into the brain, or from the ventricles, or microglia migrating from parenchyma into and around the developing blood vessels in the MNTB? It looks like that the density of the Iba1 positive cells in the ventral SOC decreases from P5 to P10. Could these cells be migrating inside the SOC and into the MNTB? Include a discussion in the manuscript.

Response: Thank you for the comment suggestion. Our study has identified cell proliferation as one source of changes in the numbers of microglia. The most direct way to determine if cell migration plays a role would be by doing time-lapse analyses in ex vivo or in vivo preparations, which is beyond the main scope of our study. However, we acknowledge that dynamic analysis of microglia is a limitation of our study, and indicate that future studies may address this topic, including a relevant reference of similar studies. See third paragraph lines 435-438 of the discussion.

4) Figure 5A-C is repetitive information of what has been reported in Figure 3. Either remove this data or consolidate it with Figure 3.

Response: The major difference between Figure 3 and Figure 5 A-C is that the latter included EdU labeling. In addition, Figure 5 includes a different sample from Figure 3. This is important because it replicates and extends the result described in Figure 3. Note that there is a statistically significant difference in the microglia ratio between P1 and P7, but there is no statistically significant difference in the percent of microglia labeled with EdU between P1 and P7.

5) As per data reported in Figure 5C, it doesn’t support the interpretation that microglia are the main perivascular cells with proliferative capacity inside the MNTB region as indicated in line 383 under discussion. In addition, the % of IBA1 and EdU double positive cells declines from P1 to P7. Perform double labeling for pericytes (NG2) or astrocytes or oligodendrocytes (GFAP or S-100ß) and EdU to determine if pericytes or astrocytes/oligodendrocytes are the main perivascular cells undergoing proliferation.

Response: We agree with the reviewer that microglia are not the main proliferative perivascular cell in this brain region during development. See the updated third paragraph lines 431-434 of the discussion.

6) Figure 7A, immunolabeling of IBA-1 doesn’t match with previous figures. Please provide a representative figure. In addition, there doesn’t seem to be any significant overlap or co-localization of either Hif1a or Vegfa in IBA1 labeled cells. At least, it not evident from the representative images provided. Rather, it is evident in figures 7 E-K, that Hif1a is abundant in NeuN positive principal neurons of MNTB while Vegfa is abundant in S-100ß positive astrocytes/oligodendrocytes.

Response: Note that the thickness of tissues used in RNAscope experiments is 20 microns, while the thickness of tissue sections used in other figures is 60-80 microns. Note that the quantification of particles per cell corresponds with the abundance of label presented in the fluorescence panels.

7) It will be worth discussing in the manuscript about the known roles of macrophages/microglia in vasculature development and angiogenesis in the brain and other tissues.  

Response: We have updated the discussion and added a reference to a relevant review on this topic. See second paragraph lines 428- 431 of the discussion.

Minor concerns

  1. Line 39: it is an incomplete sentence.

Reply: The sentence in line 39 is complete.

  1. It is unclear whether both male and female or either male or female Wistar rats were used in the study?

Reply: Note that 52 pups from either sex were used in this study (line 89).

Round 2

Reviewer 1 Report

Authors have taken care of all the suggestions and comments.

All the best.

Author Response

Thank you.

Reviewer 2 Report

Dear Editor,

The authors have successfully addressed the majority of my comments and concerns in order to improve the quality of the manuscript. 

I do believe that the corrections, new sections and updated references, have contributed to enhancing the clarity of the manuscript, which I can now endorse here.

All the best!

Author Response

Thank you.

Reviewer 3 Report

The Authors have addressed all the concerns.